# Learning in Markov Games: can we exploit a general-sum opponent?

**Giorgia Ramponi**[1]                    **Marcello Restelli**[2]

[1]Computer Science Department, ETH AI Center, Zurich, Switzerland
[2]Computer Science Department, Politecnico di Milano, Milan, Italy

## Abstract

In this paper, we study the learning problem in two-player general-sum Markov Games. We consider the online setting where we control a single player, playing against an arbitrary opponent to minimize the regret. Previous works only consider the zero-sum Markov Games setting, in which the two agents are completely adversarial. However, in some cases, the two agents may have different reward functions without having conflicting objectives. This involves a stronger notion of regret than the one used in previous works. This class of games, called general-sum Markov Games is far to be well understood and studied.

We show that the new regret minimization problem is significantly harder than in standard Markov Decision Processes and zero-sum Markov Games. To do this, we derive a lower bound on the expected regret of any "good" learning strategy which shows the constant dependencies with the number of deterministic policies, which is not present in zero-sum Markov Games and Markov Decision Processes. Then we propose a novel optimistic algorithm that nearly matches the proposed lower bound. Proving these results requires overcoming several new challenges that are not present in Markov Decision Processes or zero-sum Markov Games.

## 1 INTRODUCTION

Reinforcement Learning (RL) [Sutton and Barto, 2018] is an area of Machine Learning that studies sequential decision-making problems, where a learning agent interacts with an unknown environment to maximize its rewards. In recent years, RL methods have made substantial progress in solving real-world problems (e.g., beating the world champion player of Go [Silver et al., 2017], solving real-time strategy games [Brockman et al., 2016] and Poker [Moravčík et al., 2017, Brown and Sandholm, 2018], in autonomous driving [Shalev-Shwartz et al., 2016], learning communications and emergent behaviours [Foerster et al., 2016, Lowe et al., 2017, Bansal et al., 2018], providing solutions to robotic control problems [Lillicrap et al., 2015], and managing the power consumption of households [Chung et al., 2020]). All of these challenging real-world problems can be framed in a Multi-Agent RL (MARL) context. In Multi-Agent Reinforcement Learning (MARL), multiple agents act in the same environment, to optimize their objectives. However, despite the empirical success of MARL algorithms, theoretical understanding of MARL is relatively rare.

The MARL framework is usually modeled as a Markov Game (MG) [Shapley, 1953], which is an extension of Markov Decision Processes (MDPs) [Sutton et al., 1998]. In general, learning in MGs is harder than learning in MDPs. The complications arise from the fact that all agents affect both the transitions and the rewards of the other agents, while the agents can have completely different, even conflicting objectives. Moreover, the agents without knowledge of the transition model, have to estimate it by interactions, as in single-agent RL problems, but they need also to infer and learn the other agent's policy. In the literature [Xie et al., 2020, Wei et al., 2017, Tian et al., 2020], the learning problem in MGs has been divided into two settings: *online* and *offline*. In the *online* setting, the algorithm has the control of only one agent, to maximize its rewards in a multi-agent environment. On the other hand, the *offline* setting aims at providing self-play algorithms, i.e., algorithms that have the control of all the agents, or at least, it is assumed that all the agents use the same algorithm. While the *offline* setting has received considerable attention, it fails at modeling many use-cases of practical interest. For example, in many robot control problems, artificial agents interact with humans who are not-controllable agents; or in card/video games it is unrealistic for the opponent to use the same learning algorithm as our agent. While the online setting is more suitable to

*Accepted for the 38*[th] *Conference on Uncertainty in Artificial Intelligence* (UAI 2022).

model previous examples, it remains less studied and understood. Moreover, the algorithm that takes into account the *online* setting usually assumes that the non-controllable [1] agent has a conflicting objective, i.e. the problem that we are facing is a zero-sum Markov Game. However, there are many real-world problems where the other agent can have only different objectives, but not completely adversarial.

In this paper, we consider the *online* learning problem in a two-player general-sum turn-based Markov Game. More precisely, we consider the problem of learning in Markov Games, where there is one agent that we can control and it can observe the interaction between the non-controllable agent and the environment. We are interested in solving the following question left open in the literature:

*"Can we design a provably efficient algorithm for Markov Games exploiting a general-sum opponent?"*

This problem was also raised out in [Xie et al., 2020] as an interesting open direction. In that paper, the authors present the problem of learning in the presence of a *weak opponent*, i.e. we are facing an opponent that is not totally adversarial (as in zero-sum games). As suggested by the authors, in this case, the guarantee involves a stronger notion of regret with respect to the minimax ones.

In this paper we answer *partially* affirmative. In fact, we show that the regret minimization problem in this context is more complex than in standard MDPs. Our contributions can be summarized as follows:

1. First, we define a stronger regret definition for the problem since the minimax one [Xie et al., 2020, Tian et al., 2020] does not capture the nature of the interaction.

2. Second, we provide a novel lower bound for the regret minimization problem, that shows how the exploration problem in general-sum MGs is harder than in MDPs and in zero-sum MGs.

3. Finally, we provide an algorithm, called *Turn-based Markov Game OPtimistic Value Iteration* (TMG-OPVI) which nearly matches the proposed lower bound.

**Paper outline** The paper is structured as follows. We start by over-viewing some related works (Section 2). Then we introduce the notation (Section 3) and we provide in Section 4 a formal introduction to the problem. In Section 5, we derive a lower bound on the expected regret of any "good" learning strategy that captures the exploration challenges in this context. In particular, the lower bound clearly shows that the regret minimization in Stochastic Games is significantly more complex than in standard MDPs. Then, in Section 6, we propose an algorithm that nearly matches the proposed lower bound.

---

[1]With non-controllable we mean the policy of the second agent can not be decided by our algorithm.

**Remark 1.1.** *The considered setting is easier than the simultaneous MG setting and the one in which the agent can change its type over time. We decided to consider this framework for two reasons: 1) it is similar to the online learning zero-sum MG considered in the literature, Xie et al. [2020], and, most important, 2) we proved that even in this setting, which is a special case of the simultaneous one and of the one considered in Sessa et al. [2020], the learning problem is hard.*

## 2 RELATED WORKS

After the introduction of the concept of Markov Games [Shapley, 1953], many RL algorithms were proposed to learn in this setting. However, the theoretical study of this context is quite poor, compared to the empirical one (see the survey [Zhang et al., 2019, Da Silva and Costa, 2019, Hernandez-Leal et al., 2019, Papoudakis et al., 2019]). Only recently, there has been a growing interest in providing algorithms with strong sample-complexity and regret guarantees for the two theoretical MARL settings: *offline* and *online*. In the *online* setting, the algorithm controls only one agent, which has to maximize its own reward function. Instead, in the *offline* setting, the algorithms control all the agents in the MG.

**Offline setting** Most recent works provide results in the zero-sum offline setting, where both model-free [Bai et al., 2020, Zhang et al., 2020b] and model-based [Bai and Jin, 2020, Sidford et al., 2020, Li et al., 2020, Liu et al., 2020, Zhang et al., 2020a] algorithms were proposed with near-optimal sample complexity and regret guarantees. For the model-based setting, the prevalent approach is to assume to have access to a generative model, such as in [Sidford et al., 2020, Zhang et al., 2020a], where the authors provide non-asymptotic results on the number of queries to the generator. However, in [Liu et al., 2020] the authors proposed a model-based algorithm for the zero-sum setting without access to a generative model and which matches the information-theoretic lower bound. Furthermore, in this recent work, the authors also proposed the first line of provably sample-efficient algorithms for multi-player general-sum games. In [Li et al., 2020] the authors introduced, instead, an algorithm to learn a Nash Equilibrium in the multi-player general-sum setting. Very recently, the first algorithm to deal with sample complexity in the general-sum games (not Markov) that achieves an $\epsilon$-Stackelberg Equilibrium was introduced [Bai et al., 2021]. In this paper, the authors consider the *bandit feedback* setting i.e., they can see only the random samples of the rewards received by the two players. The authors identify a fundamental gap between the exact value of the Stackelberg equilibrium and its estimated version using finite samples. This result gives insights into the hardness of learning in General-sum games even when the setting is stateless and the algorithm has the control of both the leader and the follower.

**Online setting** The online setting is only studied, as far as we know, in the zero-sum setting. The first work that analyzes the problem of online learning in Stochastic Games is [Brafman and Tennenholtz, 2002]. In this paper, the authors propose the famous R-MAX algorithm that deals with the zero-sum average-reward setting and provides the first regret bound for the setting. In Wei et al. [2017] the authors provide an algorithm for zero-sum Stochastic games that extends UCRL2, but which works under strong reachability assumptions. This algorithm significantly improves the regret bound of R-MAX. Xie et al. [2020] propose an algorithm with a "weak" regret notion (the minimax defined before), which is compatible with a zero-sum game. This is the first work that considers linear function approximation in Markov Games. The authors analyze both the *offline* and *online* settings and their algorithms achieve near-optimal regret bounds. Instead, Tian et al. [2020] introduce the online setting with bandit-feedback, called also *agnostic* setting. In this case, the agent cannot observe any interaction between the other agent and the environment. The authors extend the method of Bai et al. [2020] to deal with this setting. In [Xie et al., 2020] the authors leave as open question how to construct an algorithm to achieve optimal regret exploiting a "weak opponent", i.e., an opponent who is not totally adversarial (as in zero-sum games). In our work, we give the first solution to this problem.

**Adversarial MDPs** The adversarial MDP problem is strictly related to the Markov Game setting. Most of the works in this setting consider adversarial rewards [Even-Dar et al., 2009, Gergely Neu et al., 2010, Zimin and Neu, 2013, Rosenberg and Mansour, 2019, Jin et al., 2020, Dick et al., 2014], i.e., the presence of an opponent who can change the received rewards. This setting is substantially different from Markov Games as the opponent can affect only the rewards and not the transitions model. Other works, instead, consider also adversarial transitions [Yu and Mannor, 2009, Neu et al., 2012, Lykouris et al., 2019]. This setting is quite challenging and the algorithms to solve this problem do not provide a $\mathcal{O}\left(\sqrt{T}\right)$ regret bound. These approaches can be applied in the bandit setting where we cannot see any feedback from the other agent, with the scope of constructing an algorithm robust to these perturbations.

## 3 PRELIMINARIES

In this section, we formally describe the background that will be employed throughout the remainder of the paper.

### 3.1 TURN-BASED MARKOV GAMES: BACKGROUND AND NOTATION

We consider two-player finite-horizon Markov Game setting [Shapley, 1953, Xie et al., 2020, Bai et al., 2020] $MG =$ $(\mathcal{S}, \mathcal{A}_1, \mathcal{A}_2, \mathcal{P}, \mathcal{R}_1, \mathcal{R}_2, \mu, H)$, where $\mathcal{S}$ is the finite state space, $\mathcal{A}_1, \mathcal{A}_2$ are the finite action spaces respectively for the first and the second player, $\mathcal{P} \in \Delta_{\mathcal{S} \times \mathcal{A}_1 \times \mathcal{A}_2}^{\mathcal{S}}$ is the transition kernel, $\mathcal{R}_i : \mathcal{S} \times \mathcal{A}_1 \times \mathcal{A}_2 \to [0, 1]$ is the reward function of the $i$-th player, $\mu$ is the initial state distribution and $H$ is the horizon. In a turn-based MG at each state only one player takes an action. The state space $\mathcal{S}$ is partitioned into $\mathcal{S} = \mathcal{S}_1 \cup \mathcal{S}_2$, $\mathcal{S}_1 \cap \mathcal{S}_2 = \emptyset$, where $\mathcal{S}_i$ is the set of states where it is $i$'s turn to play. For each state $s \in \mathcal{S}$, let $I(s) \in \{1, 2\}$ be a function that indicates the current player to play. A stochastic policy for the i-th ($i \in \{1, 2\}$) player is a sequence of H functions $\pi := (\pi_h : \mathcal{S} \to \Delta_{\mathcal{A}_1})_{h \in H}$. We define as $\pi_1 = (\pi_{1,1}, \dots, \pi_{1,K})$ and $\pi_2 = (\pi_{2,1}, \dots, \pi_{2,K})$ the two sequences of policies that, respectively, are played by our agent (agent 1) and the other agent (agent 2).

**Value Functions** The value function and action-value function, given policies $\pi_1$ and $\pi_2$, are defined for agent $i \in \{1, 2\}$, for each time step $h \in [1, H]$, state $s \in \mathcal{S}$, action $a \in \mathcal{A}$, as follows:

$$V_{i,h}^{\pi_1,\pi_2}(s) = \mathbb{E}\left[\sum_{t=h}^{H} \mathcal{R}_i(s_t, a_t)|s_h = s\right],$$

$$Q_{i,h}^{\pi_1,\pi_2}(s, a) = \mathbb{E}\left[\sum_{t=h}^{H} \mathcal{R}_i(s_t, a_t)|s_h = s, a_h = a\right],$$

where the $a_t \sim \pi_{i(s_t)}$ and $s_t \sim \mathcal{P}(\cdot|s_t, a_t)$. Furthermore, we denote with $V_1^{\pi_1,\pi_2} = \mathbb{E}_{s \sim \mu}[V_{1,1}^{\pi_1,\pi_2}(s)]$ and $V_2^{\pi_1,\pi_2} = \mathbb{E}_{s \sim \mu}[V_{2,1}^{\pi_1,\pi_2}(s)]$ the expected returns for the two agents.

The interaction between the two agents proceeds in episodes, where at the beginning of each episode the agents decide which policy to play. We indicate with $K$ the number of episodes played by the two agents. The agent observes the states, the actions played by the two agents, and noisy feedback of the agents' reward functions, i.e. $\widetilde{r}_{i,h}$ sampled from a distribution with mean $\mathcal{R}_i(s_h, a_h)$.

## 4 PROBLEM STATEMENT

In this section, we introduce the online learning problem in Turn-based General-sum Markov Games. We remark that in these games, at each step $h$ the agent $I(s_h)$ has to decide the action $a_h$ to be taken and the two players receive respectively rewards $\mathcal{R}_1(s_h, a_h)$ and $\mathcal{R}_2(s_h, a_h)$; then, the system transitions to the next state $s_{h+1} \sim \mathcal{P}(\cdot|s_h, a_h)$.

The algorithm controls only agent 1. We do not know the policy $\pi_2$ that agent 2 will play at iteration $k$ as well as the reward function $\mathcal{R}_2$ that agent 2 is optimizing. The goal is to learn a sequence of policies $\pi_1 = \{\pi_{1,1}, \dots, \pi_{1,k}\}$ that minimizes the total (expected) *regret*, defined by:

$$\mathbb{E}[\text{Regret}(K)] = \sum_{k=1}^{K} V_1^{\star} - V_1^{\pi_{1,k},\pi_{2,k}}, \qquad (1)$$

where $V_1^\star$ corresponds to the benchmarks couple of policies $\pi_1^\star, \pi_2^\star$ used to compare our algorithm. In literature, the common way of defining $V_1^\star$ [Xie et al., 2020, Wei et al., 2017] is the minimax policy defined as:

$$V^{\text{minimax}} = \min_{\pi_2 \in \Pi_2} \max_{\pi_1 \in \Pi_1} V_1^{\pi_1, \pi_2}. \tag{2}$$

The two policies $\pi_1, \pi_2$ correspond to the Nash Equilibrium of the $MG = (\mathcal{S}, \mathcal{A}_1, \mathcal{A}_2, \mathcal{P}, \mathcal{R}_1, -\mathcal{R}_1, \mu, H)$, i.e. a zero-sum Markov Game where the agent 2 is minimizing agent 1's reward function. The minimax benchmark policy is suitable to account for adversarial settings, where the other agent can adversarially change its policy to maximize the regret, or when we are in a zero-sum game. Furthermore, it was shown that in some cases, it is necessary to adopt the minimax benchmark since, otherwise, the regret minimization problem would be too difficult. An example is the *agnostic* setting when agent 1 cannot observe any information regarding the interaction between agent 2 and the environment [Tian et al., 2020].

However, in general, minimax policies do not capture the nature of the general-sum setting where the agent 2 just wants to maximize its own reward function. In fact, in the latter setting, agent 1 could hope to perform better than when facing a *non-Competitive* (its reward is not the opposite of our reward function) opponent. A benchmark that better fits the general-sum setting is the Stackelberg Equilibrium of the game:

$$V_1^{\text{SE}} = \max_{\pi_1 \in \Pi_1} V_1^{\pi_1, \text{br}(\pi_1)}, \tag{3}$$

where $\text{br} : \Pi_1 \to \Pi_2$ corresponds to a function that selects a best response policy for the agent 2 to each policy of the agent 1. Although this setting can not be applied in the *agnostic* case, in many real-world scenarios, it is plausible that the agent 1 can observe the interactions between agent 2 and the environment. Moreover, in some cases, the agent 1 can observe also agent 2 rewards or it can at least recover them (for example, using IRL approaches). In these cases, it is more reasonable to use the more challenging regret notion.

Since the *uncontrollable* agent is rationale, we can easily suppose that it plays the best response, as it is already done in literature [Balcan et al., 2015, Peng et al., 2019, Sessa et al., 2020]. More formally, the agent 2, given the policy $\pi_1^i \in \Pi_1$, follows the policy $\pi_2^{*,i}$ such that:

$$\pi_2^{*,i} \in \arg\max_{\pi_2 \in \Pi_2} V_2^{\pi_1^i, \pi_2},$$

i.e., it plays its *best response*. This creates an inherently *asymmetrical* interaction: the first agent can be seen as a *leader*, who decides the policy to be played in an episode, and the second agent can be seen as a *follower*, who can see the leader's policy and adapts its response to it. So, as in the game-theory literature [Balcan et al., 2015, Peng et al., 2019, Sessa et al., 2020], we make the following assumption:

**Assumption 4.1.** *For every policy $\pi_1 \in \Pi_1$ the second uncontrollable agent will always play the same best response policy $br(\pi_1)$, where $br : \Pi_1 \to \Pi_2$. Furthermore, $br(\pi_1)$ is deterministic.*

Under this assumption the goal of our agent is well-defined and consists in finding the policy $\pi_1 \in \Pi_1$ that is optimal under the second agent's best response policy:

$$\pi_1^\star \in \arg\max_{\pi_1 \in \Pi_1} V_1^{\pi_1, br(\pi_1)}.$$

This corresponds to finding the Stackelberg Equilibrium of the game.

We remark that agent 1 does not know the policies that the second agent will play, i.e., the $br$ function is unknown [2]. From an online learning perspective, the regret that the algorithm has to minimize is defined as:

$$\mathbb{E}[\text{Regret}(K)] = KV_1^{\text{SE}} - \sum_{k=1}^{K} V_1^{\pi_{1,k}, br(\pi_{1,k})}. \tag{4}$$

**Bandit vs Turn-based MG** Obviously, this problem can be seen as solving a stochastic multiarmed bandit problem [Lattimore and Szepesvári, 2020]. In this case, the arms are the policies, and the agent at each episode receives a random realization of its expected return. So, this problem can be solved with standard bandit algorithms such as UCB1 [Auer et al., 2002]. However, as we will explain in the next section, this is not the best we can do. In fact, the regret would not scale sublinearly with the number of possible policies, as it happens with standard bandit algorithms (where the regret is $\mathcal{O}\left(\sqrt{|\Pi_1|K}\right)$). However, we show in Section 5 that the regret has a *constant* dependence on the number of policies (i.e., not multiplicative of K). In fact, we prove a lower bound on the regret and an upper bound such that the quantity $\mathcal{O}\left(\sqrt{T}\right)$ does not scale with the number of possible policies, but with a *constant* dependence on the number of possible policies.

## 5 HARDNESS OF LEARNING IN TURN-BASED TWO-PLAYER MARKOV GAMES

In this section, we provide a lower bound on the expected regret defined in Equation 4. We remark that agent 1, i.e., the one who is controlled by the algorithm, sees the actions taken by the second agent as well as its rewards.

We consider the Turn-based MG (TMG) shown in Figure 1. That is, there are $N+3$ states, $A$ actions with $N, A \in \mathbb{N}$, and

---

[2] We can also suppose that the *uncontrollable* agent will play an arbitrary mapping between its policies and the controllable agent ones. However, if we suppose it plays the best response, we need only to know the reward function, to know the $br$ function.

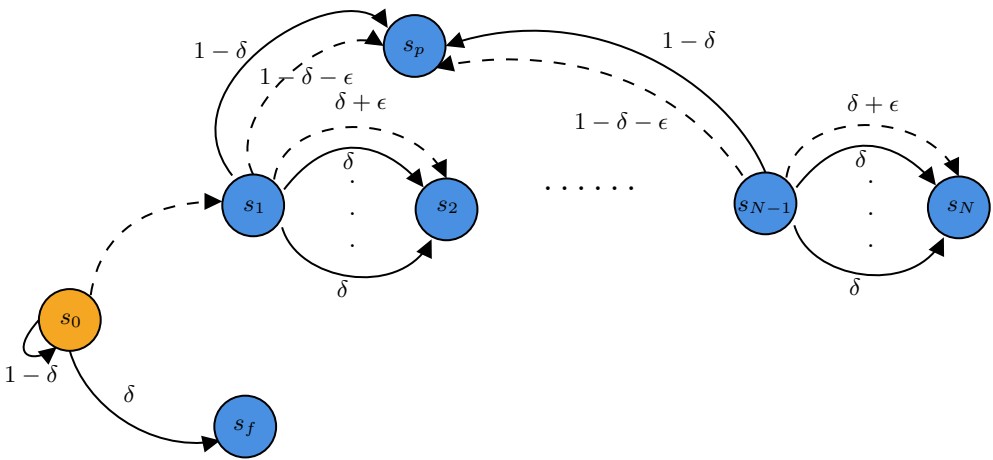

Figure 1: The composite Turn-based Markov Game was constructed for the lower bound. The states belonging to $\mathcal{S}_2$ are in orange, the ones belonging to $\mathcal{S}_1$ in blue. The dashed lines corresponds to the transition probabilities taking action $a^\star$, the others taking any other action $a \in \mathcal{A}$ with $a \neq a^\star$. The dots indicate the chain composed of $N$ states. We omit the self-loop in $s_f$, which corresponds to the fact that $s_f$ is a sink state.

$H = N+1$. The state space is defined as $\mathcal{S} = \mathcal{S}_1 \cup \mathcal{S}_2$, $\mathcal{S}_1 \cap \mathcal{S}_2 = \emptyset$. Agent 2 controls the starting state $s_0$, $\mathcal{S}_2 = \{s_0\}$, identified in the figure with the orange color. The state space of agent 1, instead, is equal to $\mathcal{S}_1 = \{s_f, s_p, s_1, \ldots, s_N\}$, i.e. it controls the blue states in Figure 1. In state $s_0$, agent 2 can choose between a *good* action $a^*$ and a *failure* action $a_f$. The reward functions of the two agents are:

$$\mathcal{R}_1(s,a) = \begin{cases} 1 & \text{if } s = s_N \\ 0 & \text{otherwise} \end{cases}$$

$$\mathcal{R}_2(s,a) = \begin{cases} R & \text{if } s = s_N \\ R_f & \text{if } s = s_f \\ 0 & \text{otherwise} \end{cases}$$

where $R, R_f \in \mathbb{R}$. The transition model of the TMG is defined as follows:

- In state $s_0$, $\mathcal{P}(s_1|s_0, a^\star) = 1$, $\mathcal{P}(s_f|s_0, a_f) = \delta$ and $\mathcal{P}(s_0|s_0, a_f) = 1 - \delta$. Then, if the agent 2 takes the *good* action, the TMG transits to state $s_1$, otherwise the next state is $s_f$ with probability $\delta$ and $s_0$ with probability $1 - \delta$.
- From state $s_f$ with any action we continue to stay in state $s_f$, i.e., $\mathcal{P}(s_f|s_f, a) = 1 \,\forall a \in \mathcal{A}$.
- From state $s_p$ with any action we continue to stay in state $s_p$, i.e., $\mathcal{P}(s_p|s_p, a) = 1 \,\forall a \in \mathcal{A}$.
- From all the other states $s_i$ with $i \in [N]$: $\mathcal{P}(s_{i+1}|s_i, a^\star) = \delta + \epsilon$ and $\mathcal{P}(s_p|s_i, a^\star) = 1 - \delta - \epsilon$; instead, for any other action $a \in \mathcal{A}$, $\mathcal{P}(s_{i+1}|s_i, a) = \delta$ and $\mathcal{P}(s_p|s_i, a) = 1 - \delta$.

The second agent has only two response functions: $\pi_{2,i}(s_0) = a_0$ or $\pi_{2,i}(s_0) = a_f$ with $i \in [1, H]$.[3] Obvi-

---

[3]It is easy to see that all other policies are not optimal, as

ously, it depends on the policy that agent 1 decides to take at the beginning of the episode. In the next proposition we prove that always exist two values for $R$ and $R_f$ such that the only policies of agent 1 that induce the second agent to take action $a^*$ are the ones such that $\pi_{1,i}(a^*|s_i) = 1$ for all $i \in [2, H]$. We call this set of policies $\Pi_1^*$.

**Proposition 5.1.** *For every* $\delta, \epsilon \in (0,1)$, $H > 3$*, there exist two values* $R$ *and* $R_f$ *such that agent 2 will play action* $a^*$ *only if agent 1 plays a policy* $\pi \in \Pi_1^*$.

**Intuition on lower bound**  From this construction, we show that agent 2 has the power to hide part of the MG. In fact, in all cases where agent 1 plays a policy different from the optimal one, we cannot acquire any further information about the transition model since we only visit state $s_f$ and $s_0$ until the end of the episode. Intuitively, we can notice that, in the worst case, we have to play all the policies in $\Pi_1$ before finding the policy that leads us to acquire information about the states in the chain. In fact, only with a policy $\pi \in \Pi_1^*$ the agent 2 allows us to visit states other than $s_f$. In the following theorem, we will formally prove this intuition.

**Theorem 5.1** (Lower bound for online Turn-based Stochastic Game)**.** *Let* $\mathfrak{A}$ *be a "good" learning algorithm, where with "good" we indicate an algorithm such that its expected regret is upper bounded by* $\mathcal{O}(CK^\alpha)$ *with* $\alpha < 1$ *in all Turn-based Markov Games* [4]*. Then we can create a Turn-based Markov Game such that the expected regret is lower*

---

either it prefers to gain $R_f$ or $R$. More details on it are provided in Appendix **??**.

[4]We note that algorithms that satisfy this assumption exist. For instance, applying UCB over the set of policies $\Pi_1$ yields regret $\mathcal{O}\left(\sqrt{|\Pi_1|K}\right)$

*bounded by:*

$$\mathbb{E}[Regret^{\mathfrak{A}}(K)] \geq \Omega\left(H\sqrt{SAK}\right). \qquad (5)$$

*Furthermore, we can create a Turn-based Markov Game with $S$ states, $A$ actions and horizon $H = S - 1$, and a specific initial distribution $\mu$ such that the expected regret of $\mathfrak{A}$ after $K$ steps:*

$$\mathbb{E}[Regret^{\mathfrak{A}}(K)] \geq \Omega\left(A^{HS}\right). \qquad (6)$$

## 5.1 PROOF SKETCH OF THEOREM 5.1

In this part, we provide a sketch of the proof of the lower bound for the online Turn-based Markov Game problem. The complete proof can be found in Supplementary **??**. We start by proving Equation 5. We can easily notice that a Markov Decision Process is a special case of a Turn-based Markov Game, in which the state space of the second agent $\mathcal{S}_2 = \emptyset$. Then from this consideration, we can state that the worst-case lower bound for MDPs can also be applied for TMGs [Jaksch et al., 2010, Domingues et al., 2020]:

$$\mathbb{E}[\text{Regret}^{\mathfrak{A}}(K)] \geq \Omega\left(H\sqrt{SAK}\right).$$

To prove Equation 6, instead, we rely on standard information-theoretic arguments used to prove lower bounds in episodic MDPs and bandit problems. More precisely, we use the following lemma due to Simchowitz and Jamieson [2019] (lemma H.1).

**Lemma 5.1** (Lower bound for online Turn-based Stochastic Game). *Let $\mathcal{TMG} = (\mathcal{S}, \mathcal{A}, H, \mathcal{R}, \mu, \mathcal{P})$ and $\mathcal{TMG}' = (\mathcal{S}, \mathcal{A}, H, \mathcal{R}, \mu, \mathcal{P}')$ be two TMGs with the same state space $\mathcal{S}$, action space $\mathcal{A}$, initial state distribution $\mu$ and horizon $H$. Fix a number of episodes $K \geq 1$ and let $\mathcal{F}_K$ be the filtration generated by all rollouts up to episode $K$. Then for any $\mathcal{F}_K$-measurable random variable $Z \in [0, 1]$,*

$$\sum_{s,a} \mathbb{E}^{\mathfrak{A}}_{\mathcal{TMG}}[N_K(s,a)] \, KL(\mathcal{P}(\cdot|s,a), \mathcal{P}'(\cdot|s,a)) \geq$$

$$kl(\mathbb{E}^{\mathfrak{A}}_{\mathcal{TMG}}[Z], \mathbb{E}^{\mathfrak{A}}_{\mathcal{TMG}'}[Z])$$

*where $kl(x,y) = x \log\left(\frac{x}{y}\right) + (1-x)\log\left(\frac{1-x}{1-y}\right)$ is the binary KL-divergence and $KL(\cdot, \cdot)$ denotes the KL-divergence between two probability laws and $N_k(s,a)$ is the number of times the state-action pair $(s,a)$ is visited till iteration $k$.*

Then we construct an alternative $MG'$ such that $MG'$ coincides with the $MG$ except in the transition from $s_0$ to $s_f$:

$$\mathcal{P}(s_f|s_0, a_f) = \delta + \epsilon \quad \mathcal{P}(s_0|s_0, a_f) = 1 - \delta - \epsilon.$$

Changing this transition the agent 2 will always play the action $a_f$. Knowing that the two games differ only in this

transition, by construction, the sum of Lemma 5.1 reduces to consider in the left-hand side only the pair state $s_0$ and action $a_f$. From this consideration, we create two events such that, the KL divergence is greater than $\mathcal{O}\left(A^{SH}\right)$.

## 5.2 DISCUSSION ON THEOREM 5.1

Theorem 5.1 shows that learning in general-sum MGs is exponentially harder than learning in MDPs. This result proves that when we are not in control of the environment it is hard to explore it in a smart way. Furthermore, we would like to remark that the setting we are analyzing also supposes a strong assumption about the not-controllable agent behavior: it can answer only with the same deterministic optimal policy. However, by removing this assumption, i.e., assuming agent 2 can choose any optimal policy, our result continues to apply.

The proof of the lower bound implicitly says that we can create very small sub-optimality gaps for the second agent and that the regret must scale with the inverse of them regardless of the suboptimality gaps of the first agent. Although we do not explicitly show this, it is intuitive to see why it can happen. In fact, agent 1 does not pay for the small suboptimality gap of the not-controllable agent but for its gap that can be potentially very high. We leave as future work to prove a problem-dependent lower bound for TMGs.

This lower bound is the first one that states the difficulties in learning in general-sum Markov Games with the possibility to see the other agent's reward function and actions. Other lower bounds were derived for the general-sum setting. In [Bai et al., 2020] the authors proposed a lower bound to underline the difficulties to learn against an adversarial opponent. In [Tian et al., 2020], instead, the authors show the statistical hardness of learning with only bandit feedback, i.e., in an *agnostic* setting. However, these two settings are harder than the one proposed in this section, and, for this reason, the results cannot be applied.

## 6 TMG OPTIMISTIC POLICIES VALUE ITERATION

In this section, we propose an algorithm, called *Turn-based Markov Game Optimistic Policies Value Iteration* (TMG-OPVI), that nearly matches the lower bound proposed in the previous section. We assume that $\Pi_1$ is any set of policies (similarly to Abbasi-Yadkori et al. [2013]), not necessarily corresponding to the full set of all deterministic Markov policies, and let $M$ be the cardinality of the policy set $\Pi_1$.

**TMG-OPVI algorithm** TMG-OPVI is a variant of Optimistic Value Iteration Azar et al. [2013], an optimistic regret minimization algorithm for finite-horizon MDPs. The algorithm proceeds as follows. Given the set of policies $\Pi_1$ for

the first agent, it stores a table recording the policy played by the second player. For every $i \in [M]$, $k \in [K]$, $h \in [H]$ we denote with $\mathcal{A}_{k,h}^i(s) \subseteq \mathcal{A}$ the set of plausible actions, i.e., the set of actions that can be played by the second agent, in state $s$ at step $h$ for policy $\pi_i \in \Pi_1$ at the beginning of episode $k$. Since, given agent 1 policy, the response policy of the other agent is deterministic and unique for assumption (see Section 3), when we play the policy $\pi_i$ and we observe in state $s \in \mathcal{S}_2$, at time step $h$, the policy $\pi_{2,h}(s)$, we can set $\mathcal{A}_{k,h}^i(s) = \{\pi_{2,h}(s)\}$.

As common in optimistic value iteration algorithms [Azar et al., 2013], we shall build upper confidence bounds to the value function of each policy by adding bonus terms based on confidence intervals on the rewards and transition probabilities. Formally, for every $k \in [K]$, state $s \in \mathcal{S}$ and action $a \in \mathcal{A}$ we derive the bonus term, based on Hoeffding's concentration inequality, for the reward function and the expected value function:

$$b_k^r(s,a) = \sqrt{\frac{2\log\left(\frac{4SAHk}{\delta}\right)}{N_k(s,a)}},$$

$$b_k^{\mathcal{P}}(s,a) = H\sqrt{\frac{2S\log\left(\frac{4SAHk}{\delta}\right)}{N_k(s,a)}}.$$

Furthermore, we indicate with $\widehat{\mathcal{R}}_{1,k}(s,a)$ and $\widehat{\mathcal{P}}_{1,k}(s'|s,a)$ the sample means of respectively the observed rewards and transitions up to (and not including) episode $k$.

Based on this, at the beginning of each episode $k \in [K]$ the algorithm computes for each policy $\pi_1^i$ with $i \in [M]$ an optimistic approximation $\widetilde{V}_{1,k}^i$ of the expected return $V_1^i$. Recursive for each $h \in [H]$, $s \in \mathcal{S}$ the optimistic approximation $\widetilde{V}_{1,k,h}^i(s)$ of the value function $V_{1,h}^i(s)$ is equal to:

$$\widetilde{V}_{1,k,h}^i(s) = \widehat{\mathcal{R}}(s, \pi_{1,h}^i(s)) + \sum_{s' \in \mathcal{S}} \widehat{\mathcal{P}}(s'|s, \pi_{1,h}^i(s))$$
$$\times \widetilde{V}_{1,k,h+1}^i(s') + b_k(s, \pi_{1,h}^i(s)),$$

if $I(s) = 1$ and, otherwise, is equal to:

$$\widetilde{V}_{1,k,h}^i(s) = \max_{a \in \mathcal{A}_{k,h}^i(s)} \widehat{\mathcal{R}}(s, a) + \sum_{s' \in \mathcal{S}} \widehat{\mathcal{P}}(s'|s, a)$$
$$\times \widetilde{V}_{1,k,h+1}^i(s') + b_k(s, a),$$

where $b_k(s, \pi_{1,h}^i(s)) = b_k^r(s, \pi_{1,h}^i(s)) + b_k^{\mathcal{P}}(s, \pi_{1,h}^i(s))$.

**Two levels of optimism**  Note that we use two levels of optimism: one for the unknown transition probabilities and rewards, and one for the unknown actions of the second agent. More precisely, if we have already seen the action that the second agent will play in a state $s$ with a policy $\pi_1^i$ we use this information to estimate the value function, otherwise we act optimistically by taking the maximum overall plausible actions. The pseudocode of TMG-OPVI is reported in Algorithm 1.

---

**Algorithm 1** TMG-OPVI

1: **Input:** $\mathcal{S}, \mathcal{A}, H, \Pi_1 = \{\pi_1^1, \ldots, \pi_1^M\}$
2: Initialize $\mathcal{A}_{1,h}^i(s) = \mathcal{A}$ for all $s \in \mathcal{S}$, $h \in [H]$, and $i \in [M]$
3: **for** episodes $1, 2, \ldots, K$ **do**
4:    Compute $\widetilde{V}_{1,k}^i$ for all $i \in [M]$
5:    Play $\pi_1^{I_k}$ with $I_k \in \arg\max_{i \in [M]} \widetilde{V}_{1,k}^i$
6:    Observe $(s_{k,1}, a_{k,1}, \ldots, s_{k,H-1}, a_{k,H-1}, s_{k,H})$
7:    Compute the plausible actions for all $s \in \mathcal{S}$, $h \in [H]$ and $i \in [M]$:

$$\mathcal{A}_{k+1,h}^i(s) = \begin{cases} \{a_{k,h}\} & \text{if } i = I_k \text{ and } s = s_{k,h} \\ \mathcal{A}_{k,h}^i(s) & \text{otherwise} \end{cases}$$

8: **end for**

---

## 6.1 REGRET GUARANTEES

In this section, we give a regret bound for the proposed algorithm. The result exploits the determinism of the other agent's policies in order to match the lower bound derived in the previous section.

**Estimation of the transition model**  The main idea behind the proof is that after having played a certain number of times every policy, we know in every state that is reachable what action the agent 2 will play. At this point, we have reduced our problem to an MDP. In fact, when we know the best response function of agent 2, for every policy we can create a policy that is the union of the policy of agent 1 and agent 2. At this point, the uncertainty comes only from the transition model and the reward function. It is important to note that we do not need to know explicitly the set of reachable states, but the algorithm implicitly will estimate correctly the agent 2 policy after having visited all of them.

Before stating the result we need to introduce some quantities. We define $d^i(s)$ as the probability of visiting state $s$ playing policies $\pi_1^i$ and $\text{br}(\pi_1^i)$. Then, we define the set

$$S_{2,h}^{+,i} = \{s \in \mathcal{S}_2 \text{ such that } d_h^i(s) > 0\}.$$

We define as $d = \min_{i \in [M]} \min_{h \in [H]} \min s \in S^{+,i} d_h^i(s)$, i.e., the minimum probability of visiting a "reachable" state. In the following theorem we provide an upper bound of the regret of TMG-OPVI algorithm.

**Theorem 6.1** (Regret of TMG-OPVI). *Let* $TMG = (\mathcal{S}, \mathcal{A}, \mathcal{P}, \mu, \mathcal{R}_1, \mathcal{R}_2, H)$ *with* $\mathcal{S} = \mathcal{S}_1 \cup \mathcal{S}_2$ *and* $\mathcal{S}_1 \cap \mathcal{S}_2 = \emptyset$ *be the finite-horizon TMG of our problem. Then the expected regret of TMG-OPVI at every episode* $K > 0$ *is bounded by:*

$$\mathbb{E}[Regret(K)] \leq \mathcal{O}\left(MSH\overline{K} + SH\sqrt{AHK\log\left(SAK^2H\right)}\right),$$

*where* $\bar{K}$ *is the first integer such that* $\overline{K} > \frac{\log\left(MS\overline{K}^2\right)}{-\log(1-d)}$

## 6.2 PROOF SKETCH OF THEOREM 6.1

In this section, we provide a proof sketch of Theorem 6.1. The complete proof can be found in Appendix **??**. We consider the fact that, at some point, if we play a certain policy $\pi_1$ we will observe every state that can be reached playing $\pi_1^i$ and $\text{br}(\pi_1^i)$. In fact, there exists an iteration $\overline{K}$ such that each state $s \in S_{2,h}^{+,i}$ with $i \in [M]$ has been visited at least one time. After $\overline{K}$, agent 1 has complete knowledge of the best response function $\text{br}$[5]. From this iteration, the algorithm is facing a single-agent problem, where the joint policy is derived by the union of the policies of the two agents:

$$
\pi^i(s) = \begin{cases} \pi_1^i(s) & \text{if} \quad I(s) = 1 \\ \mathcal{A}_{k,h}^i(s) & \text{if} \quad I(s) = 2 \end{cases},
$$

where in this case $\mathcal{A}_{k,h}^i(s)$ is a singleton for every state $s \in \mathcal{S}_2$ and policy $\pi_1^i$ with $i \in [M]$. Then we can proceed with our proof considering this new single-agent problem.

## 6.3 DISCUSSION ON THEOREM 6.1

The following regret nearly matches the proposed lower bound. In fact, if we instantiate the set of policies of agent 1 equal to all the possible deterministic policies, then $M = A^{HS}$ where $A$ is the cardinality of the action space and $S$ the cardinality of the state space and $H$ is the horizon. Instead, the second term of the regret is comparable with the worst-case lower bound for MDPs [Azar et al., 2013].

It is interesting to note that respect to *agnostic* MG [Tian et al., 2020] we achieve better regret guarantees in terms of $K$ in our setting where they achieve a regret upper bound of $\mathcal{O}\left(K^{\frac{2}{3}}\right)$. We achieve also stronger regret guarantees with respect to adversarial MDPs [Abbasi-Yadkori et al., 2013] where transitions and rewards can change adversarially. The two settings are quite similar, since also in the setting considered in this paper the transitions change *adversarially*, since agent 2 influences the transitions of the MG. However, in this case, differently to ours, the regret that is achieved is $\mathcal{O}\left(\sqrt{K \log(M)}\right)$ while our regret does not depend on the number of policies in the $K$ term. On the other hand, their constant dependence on the number of policies is $\mathcal{O}(\log(M))$, while we obtain a constant dependence on the number of policies. Clearly, the two settings, are different, so it is hard to compare them, but our result shows that using the information of the not-controllable agent, we can achieve better performances.

---

[5]It is important to notice that we do not have to know the sets of reachable states, but we use these sets only for a proof purpose.

## 7 CONCLUSIONS

**Contributions** In this paper, we propose the first insights to the online learning problem in general-sum Stochastic Games. Although there are some recent results in solving the problem in the zero-sum (aka competitive) setting, there are no other works that take into account the problem or consider that we could face a non-Competitive opponent. We have shown that the problem is much more complicated than in a zero-sum MG and an MDP The main problems arise from the limited control on the environment's exploration. We underline this difficulty by providing a novel lower bound (Section 5), which proves that the regret scales constantly with the number of deterministic policies that can be played by the controllable agent. This creates a big gap between what we can obtain learning in MDPs with respect to general-sum MGs. Then we show how to build a provably efficient algorithm in Section 6. Our algorithm, TMG-OPVI, achieves optimal performance nearly matching the proposed lower bound. We would like to underline that this is the first paper that considers the online learning problem in the general-sum Markov Game, and we think that our findings help in the understanding of the MARL problem.

**Future directions and discussion on learning in Markov Games** Currently, there is a need for a formal understanding of the online MARL problem to construct provably efficient learning algorithms for this context. As our result suggested, the MARL setting poses novel challenges, especially in the well-known exploration-exploitation dilemma, i.e., the trade-off between gathering new information and exploiting it: in a multi-agent environment, the agent needs to explore not only to understand the underlying environment but also to learn the other agents' behaviors. Moreover, from our findings, it is clear that the algorithm design and the resulting performance guarantees heavily depend on any knowledge about the opponents, either known as a priori or obtainable during the learning process.

Furthermore, there are open problems also in the agnostic setting, presented in [Tian et al., 2020], as it is possible to achieve better theoretical (regret) guarantees and construct algorithms with optimal sample complexity. This scenario, having no assumptions on the opponents, is widely applicable to capture real-world problems. On the other hand, assuming to have the possibility to observe other agents' interactions with the environment or having some previous knowledge about the other agents (as having access to a finite set of opponents [Balcan et al., 2015] or considering a larger set of opponents' classes with some regularity assumptions [Sessa et al., 2020]) we could hope to obtain better theoretical guarantees. An unexplored, but promising future direction would be considering some structural relation between the best response of the agent 2 and agent 1. It could overcome the problem of the constant dependence on the number of deterministic policies.

Another interesting future direction is to better show the relationship between the sub-optimality gap of the non-controllable agent and the one of the controllable agent. To prove this it is necessary to prove a problem-dependent lower bound. We leave this analysis as future work.

## Acknowledgements

We thank Andrea Tirinzoni for valuable discussions.

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
