# OpenReview forum: "Learning in Markov Games: can we exploit a general-sum opponent?"
_auai.org/UAI/2022/Conference — UAI 2022 Oral_

### Official Review · Reviewer_NKn8 · 2022-03-16

**Q2(1) Originality/Novelty:** 3
**Q2(2) Significance/Impact:** 3
**Q2(3) Correctness/Technical Quality:** 2
**Q2(6) Clarity Of Writing:** 3
**Q6 Overall Score:** 6
**Q8 Confidence In Your Score:** 3

**Q1 Summary And Contributions:**

This paper presents a lower bound for learning complexity general-sum Markov games, specifically for the Stackelberg equilibrium concept. Additionally, an “optimism in the face of uncertainty”-style algorithm is proposed which comes close to attaining the theoretical lower bound.


**Q2 Assessment Of The Paper:**

More detailed information regarding each of these aspects is given below:

**Q2(4) Quality Of Experiments (Optional):**

1: Poor: The experimental evaluation is flawed or the results fail to adequately support the main claims.

**Q2(5) Reproducibility:**

3: Good: Key resources (e.g., proofs, code, data) are available and key details (e.g., proofs, experimental setup) are sufficiently well-described for competent researchers to confidently reproduce the main results.

**Q3 Main Strengths:**

- Presentation is clear, contributions are well-motivated and connected with one another and the relevant literature.
- Theoretical lower bound appears to be novel, and to pertain to a large class of games which is broadly relevant.
- Likewise, algorithm to (nearly, asymptotically) attain this bound appears simple to implement and broadly applicable.

**Q4 Main Weakness:**

- Lemma 5.1 does not appear in the reference indicated so far as I see. Please explain.
- Still in the proof of theorem 5.1, it is unclear to me: (a) why the construction of figure 1 constitutes a lower bound, and (b) why one can apply a lower bound result from MDPs to the Turn-based Game setting. {Games} \includes {MDPs}, so a lower bound in the game setting should also be a lower bound for MDPs, but not the other way around, correct? Perhaps I am misreading the statement here, but in any event, I feel that the whole discussion of this proof sketch could be clarified dramatically to give insight into the major steps of the proof while gently directing the interested reader to the appendix for further details.

**Q5 Detailed Comments To The Authors:**

- The development of the Stackelberg problem here left me confused as to why we are studying this equilibrium concept instead of Nash (or something else). After all, one can certainly have both equilibrium concepts in general-sum (or zero-sum) games. I suspect that we are studying Stackelberg problems here because, they may be reduced to single-player problems as alluded to in Section 6.1. If this is indeed the primary motivation for studying this equilibrium concept, it should be stated outright.
- Solutions to (2) are not necessarily a Nash equilibrium… rather they are Stackelberg.
- N_K is never defined
- The random variable Z discussed in Lemma 5.1 is never explicitly characterized, so far as I can tell
- In line 7 of Alg. 1, should we not be accumulating sets of actions with union operators, not simply replacing them?
- Formatting of long eqn line in Theorem 6.1 goes into the margin
- Why is the subscript “p” used (and, oddly, colored red in some places)? At first I thought it was the p^th state in the top chain, but it appears to be separate. I suggest changing the notation to be clearer.
- Missing a period following “MDP” in the conclusion. Other minor syntax errors throughout - suggest a careful read.
- I love the intuition in the section “Estimation of the transition model” - it would be great to have this woven throughout the paper!

**Q7 Justification For Your Score:**

As above, I feel that this paper provides novel theoretical understanding of a difficult topic in competitive learning, and will make a significant impact on the field when it is published (here or in a future venue). However, there are a number of technical questions which I have raised above and which ought to be addressed prior to publication.


[Edit]: I feel that my comments have largely been addressed and have changed my score accordingly. Thank you very much for your clear explanations.

**Q9 Complying With Reviewing Instructions:**

1: Yes.

---

### Official Review · Reviewer_3oh4 · 2022-04-13

**Q2(1) Originality/Novelty:** 3
**Q2(2) Significance/Impact:** 3
**Q2(3) Correctness/Technical Quality:** 3
**Q2(6) Clarity Of Writing:** 3
**Q6 Overall Score:** 7
**Q8 Confidence In Your Score:** 3

**Q1 Summary And Contributions:**

This paper studies solving general-sum Markov Games. In detail, the authors assume that the opponent is rational and she executes a deterministic policy against the main player. By showing both upper and lower bounds of regret, the authors suggest that for general-sum MG, its statistical complexity depends on the cardinality of the policies the main player can take.

**Q2 Assessment Of The Paper:**

More detailed information regarding each of these aspects is given below:

**Q2(5) Reproducibility:**

3: Good: Key resources (e.g., proofs, code, data) are available and key details (e.g., proofs, experimental setup) are sufficiently well-described for competent researchers to confidently reproduce the main results.

**Q3 Main Strengths:**

+ The presentation is clear.
+ The proof sketch is easy to understand.
+ The discussions are comprehensive.


**Q4 Main Weakness:**

- The restriction of the model class makes it hard to justify the importance of this work.

**Q5 Detailed Comments To The Authors:**

- My main concern is the assumption that the opponent player will take a fixed policy against each main player’s policy. It seems that such an assumption simplifies the problem a lot and makes the proposed algorithm work (without such an assumption, the action set elimination strategy may not work). Is this assumption necessary?

**Q7 Justification For Your Score:**

This work provides the first theoretical result about general-sum MG, which suggests that the statistical complexity to learn the MG should depend on the cardinality of the policy space, which is new compared to the zero-sum MG setting. I think this work contributes to the RL research community. I recommend accepting this paper.

**Q9 Complying With Reviewing Instructions:**

1: Yes.

---

### Official Review · Reviewer_4sxt · 2022-04-13

**Q2(1) Originality/Novelty:** 3
**Q2(2) Significance/Impact:** 3
**Q2(3) Correctness/Technical Quality:** 3
**Q2(6) Clarity Of Writing:** 3
**Q6 Overall Score:** 6
**Q8 Confidence In Your Score:** 1

**Q1 Summary And Contributions:**

This paper considers two-player Markov Games  (turn-based)in the general case (the game is not necessarily zero-sum). The learning problem deals with one agent only,  the other one being assumed to play his best response. A regret criterion based on the notion of Stackelberg equilibrium is then defined. A lower bound of this expected regret is proposed and an algorithm is presented which is supposed to "nearly" optimize this lower bound.

**Q2 Assessment Of The Paper:**

More detailed information regarding each of these aspects is given below:

**Q2(4) Quality Of Experiments (Optional):**

2: Fair: The experimental evaluation is weak: important baselines are missing, or the results do not adequately support the main claims.

**Q2(5) Reproducibility:**

3: Good: Key resources (e.g., proofs, code, data) are available and key details (e.g., proofs, experimental setup) are sufficiently well-described for competent researchers to confidently reproduce the main results.

**Q3 Main Strengths:**

As far as I understand (the domain is not mine), the problem considered is original and the authors have convincing arguments in favor of its significance. I agree with the fact that games are generally not zero-sum, and that the optimization of the policy of one agent, the other being supposed to play his best response is realistic and useful.

A convincing formalization is proposed as well as a regret criterion.  A lower bound of this expected regret is proposed and an algorithm is presented which is supposed to "nearly" optimize this lower bound.

**Q4 Main Weakness:**

First of all, experiments are missing

Some points are unclear in the writing, and in particular the claim that  "the problem is hard", which motivates the proposition of a lower bound.

**Q5 Detailed Comments To The Authors:**

* The link between equation (4) and the claim that this leads to finding a Stackelberg equilibrium (as said in the previous paragraph) is not trivial for me

* I did not catch why that the problem is "Harder" (section 5).

* when does the lower bound become exact ? do we have some guarantees of the efficiency of the lower bound ?

* the algorithm is supposed to "nearly match" the lower bound -  when does it fail to match it ? what is the quality of the approximation with respect to not only the lower bound, but above all the criterion to optimize (equation 4) ?

* because there are two levels of approximation  ("nearly matching" a "lower bound"), we need a  formal or experimental evaluation of the quality of the solutions provided by the algorithm

**Q7 Justification For Your Score:**

The problem considered seems new and important, the approach is original and the formalization provided is convincing. Proofs are provided. Nevertheless, because there are two levels of approximation  the paper lacks a  formal or experimental evaluation of the quality of the solutions provided by the algorithm.  It also lacks a an experimental bassement of the temporal complexity - the algorithm is supposed to run "online" - is it quick enough in practice ,

**Q9 Complying With Reviewing Instructions:**

1: Yes.

---

### Official Review · Reviewer_okMS · 2022-04-13

**Q2(1) Originality/Novelty:** 3
**Q2(2) Significance/Impact:** 3
**Q2(3) Correctness/Technical Quality:** 3
**Q2(6) Clarity Of Writing:** 3
**Q6 Overall Score:** 7
**Q8 Confidence In Your Score:** 3

**Q1 Summary And Contributions:**

This paper studies general sum games, by providing optimistic algorithms and studying their regret guarantees.

**Q2 Assessment Of The Paper:**

More detailed information regarding each of these aspects is given below:

**Q2(4) Quality Of Experiments (Optional):**

3: Good: The experimental evaluation is adequate, and the results convincingly support the main claims.

**Q2(5) Reproducibility:**

3: Good: Key resources (e.g., proofs, code, data) are available and key details (e.g., proofs, experimental setup) are sufficiently well-described for competent researchers to confidently reproduce the main results.

**Q3 Main Strengths:**

Regret guarantees for this setup is a good contribution.

**Q4 Main Weakness:**

None noted.

**Q5 Detailed Comments To The Authors:**

See above.

**Q7 Justification For Your Score:**

The results are good.

**Q9 Complying With Reviewing Instructions:**

1: Yes.

---

### Decision · Program_Chairs · 2022-05-15

**Decision:**

Accept (Oral)

**Comment:**

Meta Review: The reviewers felt that this paper offered a valuable contribution to the analysis of learning in games and MARL, specifically through its focus on general-sum Markov games, and the analysis of the expected regret of learning agents in this setting. The results stand in contrast to those for general MDPs and zero-sum games. The reviewers did raise a number of questions and made a number of suggestions for improving the paper especially w.r.t. clarity of the presentation. The author response clarified most of the concerns of the reviewers, but it is important for the authors to ensure that their revised paper reflect these improvements.